# Study for Performance Increase of a Extractor Device by Steel Replacement of AISI 304 Steel for AISI 420 Steel

**DOI:** 10.3390/ma15010280

**Published:** 2021-12-30

**Authors:** Francisco Alves de Lima Júnior, Ricardo Artur Sanguinetti Ferreira, Rômulo Rocha de Araújo Lima

**Affiliations:** Center for Technology and Geosciences (CTG)—Campus Recife, Federal University of Pernambuco (UFPE), Avenue Professor Moraes Rego 1235, Cidade Universitária, Recife 50670-901, Brazil

**Keywords:** extractor device performance, structural analysis, AISI 304 and AISI 420 steel

## Abstract

The performance of an extractor device used in the food industry was studied from the development of structural analysis through computational modeling based on finite elements. These analyses considered the mechanical properties of AISI 304 and 420 stainless steels, in addition to the tribological aspects of the device in operation. Initially, uniaxial tensile tests were carried out according to the ABNT NBR 6892 standard and hardness tests were carried out according to ASTM E384, E92, and E18 standards. From the mechanical tests, structural analyses were carried out numerically on each of the components of the extractor device. After analyzing all the components, the device was assembled to be tested in operation. The wear and service life of devices made from these two materials were evaluated. From this study, it could be concluded that the extractor device made with AISI 420 stainless steel, in addition to having a lower manufacturing cost, suffered less wear and had an increase in service life of up to 650% compared to the extractor device made with steel stainless steel AISI 304.

## 1. Introduction

Modern or world-class industries increasingly seek to develop structured production systems with methodologies and instruments that enable widespread application throughout the organization [1].

Aiming at the application of these concepts and techniques in a manufacturing environment, demands have emerged for the development of new devices and components aimed at improving the performance of the equipment that make up the plant production lines [2].

Within a more specific approach, in the food industry, stainless steels are the materials most widely used in the construction of equipment, systems, and manufacturing processes due to their great versatility. For this reason, their employment has been playing an increasingly important role, perhaps, fundamental, in our daily lives [3].

Stainless steels appear as excellent materials for these applications as they are indicated for use in environments or equipment subject to extreme working temperatures (high or low) and atmospheres with high corrosion rates, when there is a need for non-contamination of fluids by residues of corrosion, activities involving dangerous fluids, when maximum protection against possible leaks is required, and finally for applications in equipment or components subject to great mechanical stress or that are subject to microbiological contamination [4].

However, aiming at a correct choice for the subsequent application of stainless steels in mechanical components and devices, it is extremely necessary to develop methods, processes, tools, and analyses suitable for the materials studied, so that a final product is produced that meets all the requirements of mechanical, structural resistance, and manufacturing quality [5].

This work aims to develop, in an unprecedented way, a study to verify the improvement in mechanical properties and performance of an extractor device used in the Food Industry, through the application of 420 stainless steel in replacement of 304 stainless steel, using as a basis an analysis comparison between the mechanical properties of these types of stainless steels, the development of a specific tribological system, the use of structural analysis using finite element modeling, and the execution of tests and tests of performance and quality control.

## 2. Materials and Methods

### 2.1. Materials

In order to carry out the analysis and testing of the materials, test specimens extracted from metallic elements of devices already existing and used in operation, and from others never used in operation, were used in a first stage. In a second stage, specimens extracted from metallic elements developed to compose the new devices were used. Figure 1 shows the components selected from existing devices, intended for making specimens for the analysis as well as the analysis and testing of materials.

The manufacture of the prototypes of the extractor device for the study was made in AISI 304 stainless steel and the final version in AISI 420 stainless steel (commonly recommended materials). The specifications of these materials used in manufacturing are listed in Table 1.

### 2.2. Methods

After cutting the samples, initial preparation, and identification of the specimens, they were separated according to each analysis or test to be performed, along with their respective equipment, systems, and specific instruments [6].

#### 2.2.1. Structural Analysis of Components

Structural analysis is the study and determination of loads acting on a given structure and its subcomponents [7]. Through knowledge of applied mechanics, materials science, and numerical methods, it is possible to obtain the stress distribution in the structure, given an adequate failure criterion, deformations, internal forces, accelerations, and service life, fixed to a payload.

Specifically in relation to the project, the structural analysis was carried out with the purpose of also verifying the efforts present during the device operation and which of the components are more concerning, regarding the possibility of failure occurrences in their structures. Initially, a pre-analysis of the critical components of the system as a whole was carried out, raising which parts were of more concern and which choices could be made in order to maintain the fidelity of the results [8].

In the present project, for each structure considered critical, a single analysis was carried out, in isolation, so that the component could be better examined [9]. A systemic plan was prepared to support the execution of structural analysis activities, as shown in Figure 2.

In the analysis of the device extractor assembly, an indirect boundary condition was considered, thus applying an initial displacement and not an initial force, performing a thirty-degree displacement of the extractor assembly in a clockwise direction (actual displacement of the device, verified in the field), so that it was possible to perform the simulation without having to estimate a net force, enabling a more reliable result, as shown in Figure 3.

The drive rod is the most critical component of the device, which has a considerable amount of breakage in short periods of time [10]. For this component, there were two main factors that could cause the number of excessive breaks, one being fatigue failure, where a component subject to repetitive strain can fail even with stresses below the yield limits and the other the release movement of the product, where a metal bar makes contact with the rod, which results in a very high impact load due to the short contact time between the bar and the rod of the device.

For this analysis, the force acting on the rod was considered as an impact load due to the operating conditions of the device [11]. As it is an impact between two materials with considerable hardness and tenacity, the contact time would be very small, with the amplification of the acting efforts [12].

In order to perform an equivalence, in order to be able to carry out a structural static analysis of the device’s drive rod, it is necessary to calculate the impact force based on the speed and contact time between shocks in the elastic regime of metal to metal, that is, to transform the instantaneous effort made by the impact of the drive rod with the equipment bar into a static load. For this, the impact theory of Hertz [13] was used, from experimental measurements carried out using the equipment, in the field, operating at maximum production speed.

Thus, the estimated shock or contact time can be calculated using Equation (1) [13], being expressed in milliseconds (ms). It was assumed that the bar and rod materials are the same, considering that the values of the main unknowns (elasticity modulus and Poisson’s coefficient) practically do not vary if we change the bar material to another stainless steel.
(1)$$T_{c}=6.46\;\rho^{\frac{2}{5}}\left[\frac{R}{U_{o}{}^{\frac{1}{5}}\;\;E^{\frac{2}{5}}}\right]$$

In Equation (1), *ρ* is the density or Poisson’s coefficient of the material, expressed in grams per cubic centimeter (g/cm^3^), *R* is the radius of the rod, expressed in millimeters, *U_o_* is the contact velocity, expressed in meters per second, and *E* is the modulus of elasticity of the material, expressed in kilo-newtons per square millimeter (KN/mm^2^).

The impact can be considered as the variation in the amount of movement, according to Equation (2), and it can also be expressed as the multiplication of the impact force by the impact time [13], according to Equation (3).
(2)$$I=\Delta Q\;ou\;I=mv_{1}-mv_{2}$$
(3)$$I=F_{i}\;\Delta T_{c}$$

In Equation (2), *m* is the component mass, expressed in kilograms (kg) and *v* is the impact velocity, expressed in meters per second. In Equation (3), *F_i_* is the impact force, expressed in Newtons (N), and *T_c_* is the shock or impact time, expressed in milliseconds.

Using the previous equations, we were able to find the static charge value for the effort due to the shock. With this value, it is possible to perform a static analysis, which will provide the structure’s behavior with great precision, without the computational cost that would exist if it were chosen to perform a dynamic analysis [14]. Figure 4 represents the finite element structural mesh of the drive rod and structural analysis with defined boundary conditions.

The device has, in addition to the metallic elements, a carbide insert, whose structural analysis involves more peculiar criteria and considerations. The effort to which it is subjected, despite being mechanically simple, is extremely complex to be modeled by a computational method, since the contact area between the material handled by the device and the carbide insert being variable, the applied stresses consequently are too (since they depend on the area of contact), as does the speed of movement. These three factors are decisive for a computational analysis [15].

The finite element method is a numerical method for the approximate solution of a differential equation [16]. However, the differential equation that represents the problem is extremely complex, and possibly a numerical solution does not obtain reasonable precision [17]. For engineering purposes, an analysis was performed using concepts of tribology (area destined to the study of friction between materials), in order to obtain an estimate of the number of cycles for the materials [18].

As the material handled by the device goes through the process only once, we are only interested in insert wear. Thus, it was necessary to analyze the surface wear (by abrasion) of this component.

In general, wear is inversely proportional to hardness [19]. The wear rate can be determined through a test, however, the wear volume is independent of the sliding speed and can be calculated by Equation (4) [20], being expressed in meters per second.
(4)$$V=K_{d}\left[\frac{F_{n}\;L}{H}\right]$$

In Equation (4), *K_d_* is the material wear coefficient (dimensionless), *F_n_* is the normal applied force, expressed in Newtons, *L* is the length of the slip between the insert and the material handled by the device, expressed in millimeters, and *H* is an absolute hardness of the material handled by the device (wood), expressed in Newtons, on the “Janka Hardness” scale.

However, the relevant variable for the analysis is the depth of wear. It can be calculated from Equation (4), dividing the wear volume value “*V*” by the average contact area, according to Equation (5), being expressed in millimeters, where *A_c_* is the apparent contact area between the insert and the material manipulated by the device, expressed in square millimeters (mm^2^). Unlike computational analysis, there are only linear variations, thus being able to consider the average area.
(5)$$d=K_{d}\left[\frac{F_{n}\;L}{HA_{c}}\right]$$

In the structural analysis of the base of the device, the efforts and displacements that occur in the operation of the device are transferred from the extractor assembly and the drive rod to the base through the axes that connect to the holes “*A*”, “*B*”, “*C*”, and “*D*”, according to the simulation in ANSYS Workbench 14^®^ software, as shown in Figure 5. To determine the intensity of the efforts, these holes were considered as supports, and their equilibrium reactions were calculated from the force balance equations and moments in the three coordinate directions (x, y, z).

To perform the calculations, the situation of greatest effort of the device was considered, which corresponded to the position in which the mainspring (component with structural analysis not included in this work) is at its moment of greatest traction. In addition, the strategy of dividing the device into two parts was adopted, as described below and shown in Figure 6:
Part I: Forces acting on the extractor set and their influence on supports A and B;Part II: Forces acting on the drive rod and their influence on supports C and D.

Before calculating the equilibrium reactions, it is necessary to calculate the force exerted by the mainspring, which is possible based on the knowledge of its elastic constant “*K*” and its length variation, from the “untensioned” position up to maximum traction condition. The mainspring traction force can be calculated by Equation (6) [21], being expressed in Newtons.
(6)$$F\_m=K\;\left[\Delta s\right]$$

In Equation (6), *K* is the spring’s elastic constant, expressed in Newtons per millimeter (N/mm), and ∆*s* is the variation of its length, from the “untensioned” position to the maximum tension condition, expressed in millimeters. The elastic constant of the spring can be calculated by Equation (7) [21], being expressed in Newtons per millimeter.
(7)$$K=\frac{d_{m}{}^{4}\;G}{8\;D_{e}{}^{3}N_{a}}$$

In Equation (7), *d_m_* is the diameter of the wire used in the spring, expressed in millimeters, *G* is the shear modulus (or stiffness) of the material, expressed in gigapascals (GPa), *D_e_* is the mean diameter of the turn, expressed in millimeters, and *N_a_* is the number of active turns (dimensionless). The value of Δ*s*, in turn, has already been calculated in the mainspring design.

When considering their performance in the extractor assembly, the meanings of the components are inverted, and consequently their signs are inverted. However, aiming to calculate the equilibrium equations, the value of all components was considered as positive and their respective signs were added only in the equations.

The depth of wear by abrasion on the base can be calculated by Equation (5) (Archard’s Equation), being expressed in millimeters [22]. In addition to the geometric parameters of the slip and the normal force in the contact between the base and the material handled by the device, it is necessary to understand the wear coefficient, obtained experimentally [23].

As this is an unprecedented work, there is no evidence of experimental results for a tribological system consisting of the base and the material handled by the device, requiring research to support the study, in order to find an approximate value for the wear coefficient for the materials used in the base, in interaction with other abrasives, which despite having a much greater hardness and consequent wear severity, serve for purposes of comparison between these materials used and the material handled by the device as well as having an idea of the durability of both [24].

A measure that indicates the severity of wear is the “*H_a_/H*” ratio, where H_a_ is the hardness of the abrasive, expressed in MPa. When 0.8 < *H_a_/H* < 1.2 wear is considered moderate [25]. There is a variation in Archard’s equation (Equation (5)), where a dimensional (*k*) or dimensionless (*K_d_*) wear coefficient can be used, which are related through the hardness (*H*) through Equation (8):(8)$$k=\frac{K_{d}}{H}$$

For this calculation, the hardness of the less rigid material is generally used, as it tends to wear out more quickly. However, as each material handled by the device only participates in one cycle, the hardness value of the material used in the base will be used, given the interest in calculating the wear value of this component. Figure 7 represents the finite element structural mesh of the component and structural analysis with defined boundary conditions.

#### 2.2.2. Tensile Test

The purpose of this test is to verify the tensile strength and ductility of materials when subjected to constant load until their fracture. In this test, it is also possible to verify the yield stress and maximum tensile strength of the tested materials.

In this analysis, specimens of each type of material are extracted, with specified dimensions, and each material is inspected dimensionally and visually for its physical integrity. All test parameters are then entered into software that communicates with the respective universal testing machines [26].

Finally, the specimens were uniaxially tensioned, with a constant strain rate of 0.0025 s^−1^, according to the [27], in order to stop the test when they broke, more specifically in their useful areas. Figure 8 shows the execution of tensile tests on the analyzed materials.

#### 2.2.3. Hardness Analysis

The purpose of this analysis is to verify the penetration resistance of materials, in order to learn their mechanical and wear resistance as well as the effect of the heat treatments used.

##### Vickers Hardness Analysis

In this analysis, specimens of each type of material are extracted, with specified dimensions, and then placed on the base table of a specific durometer, where they are penetrated by a diamond pyramid with a square base and an angle between sides of 136°, under a load specified of 1961 Newtons, for a time of 30 s, when the equipment was turned on, generating three results, according to the [28].

##### Hockwell Hardness Analysis

In this analysis, specimens are also extracted from evidence of each type of material, with specified dimensions. In this method, the load is applied in stages, that is, a preload is first applied through a standard block, for calibration, in order to ensure firm contact between the penetrator and the analyzed material, then the load is applied itself, according to the [29]. In these tests, five measurements were performed, discarding the lowest and highest values, in order to evaluate the values closest to the real average. Figure 9 shows the execution of the hardness analyzes of the analyzed material.

## 3. Results

### 3.1. Component Tensile Tests

The results of the uniaxial tensile tests performed on components manufactured in AISI 304 stainless steel are shown in Table 2.

The behavior of the tensile–strain curves of the components in sheet and cylindrical shapes, manufactured in AISI 304 stainless steel, obtained from the tensile tests, is shown in Figure 10.

The results of uniaxial tensile tests performed on components manufactured in AISI 420 stainless steel are shown in Table 3.

The behavior of the tensile–strain curves of the components in sheet and cylindrical shapes, manufactured in AISI 420 stainless steel, obtained from the tensile tests, is shown in Figure 11.

When comparing the types of materials analyzed, considering the same shapes of specimens, it appears that for the sheet format, specimens made of AISI 304 stainless steel had a lower tensile strength and lower tensile strength than the AISI 420 stainless steel specimens. For the cylindrical format, AISI 304 stainless steel specimens had a higher tensile strength limit compared to the AISI 420 stainless steel. As will be shown below, this result is justified by the respective manufacturing processes of plate and cylindrical shapes [30].

### 3.2. Component Hardness Analysis

The results of the hardness analyses performed on the components manufactured in AISI 304 stainless steel were obtained from the submission of materials to penetrating elements, generating the data listed in Table 4.

In light of the results obtained from the hardness analyses, it appears that the values presented by the specimens in cylindrical formats are superior to the values presented by the specimens in plate formats.

The results of the hardness analyses performed on components manufactured in AISI 420 stainless steel were obtained from the submission of materials to penetrating elements, generating the data listed in Table 5.

In light of the results obtained from the hardness analyses, it appears that the materials of the specimens in plate and cylindrical formats, made of stainless steel type AISI 420, had, as expected, values higher than the values found in the respective bodies of proof, in plate and cylindrical formats, manufactured in stainless steel type 304. This difference between plate-shaped and cylindrical elements is justified by the greater hardening of cylindrical elements, produced by drawing, in relation to plate-shaped elements, produced by lamination [31].

### 3.3. Structural Analysis of Components

The computational models used for structural analysis are presented separately, according to each studied component.

In the structural analysis of the extractor set of the devices made of AISI 304 stainless steel and 420 stainless steel, it is possible to notice that the tensions were much lower than any other component. In components made of AISI 304 stainless steel, the maximum tension requested from the element is around 5.0 MPa, while in components made of AISI 420 stainless steel, the maximum tension is around 4.56 MPa, which are practically equal values when considering the values of the material yield stress limits (about 206 MPa and 345 MPa, respectively), as shown in Figure 12.

Figure 13 shows the stress distributions in the drive rod manufactured in AISI 304 stainless steel, obtained by the Von-Mises criterion and the consequent deformations in the component, caused by the application of these stresses, respectively. Deformations are indicated in millimeters and stresses in MPa.

Observing the behavior of a sample series of worn drive rods, it was verified that the shock occurred about 25 mm from the upper tip, which causes a significant change in the applied stress values.

There are two factors that are of concern in the component structure. The first, with regard to its safety factor, is the ratio between the maximum allowable voltage and the critical value obtained in the analysis [32].

For the analysis, 225 MPa was adopted as the maximum allowable stress value, an intermediate value between the maximum and minimum, considering that the material has a reasonable quality.

Since we found, by the finite element method, the critical value of 205 MPa, we arrived at a safety factor value around 1.1, as shown in Figure 14. This means that the component will withstand up to 10% excess, in relation to the applied effort.

However, 10% is a significantly small and worryingly low value, as small differences in material quality, higher equipment operating speed, or any other factor not considered can lead to greater effort, which would lead to component failure.

By also analyzing the material’s compression limit [32], we noticed a reduction in the safety factor, causing any greater effort to lead the component to a simple static failure.

A trigger cycle is the number of times there is a load and change of its alternating component. Therefore, for the device, the number of cycles per fault will be the number of times the drive rod performs a full cycle movement. It is possible to observe that there is a clear influence on the value of the voltage level due to possible changes in the material used and in the location of the shock.

Of these, the change in the point of force application is the most significant, indicating that the component is sensitive to this change, and this factor is something that must be observed if there is a need to modify the operation of the equipment containing the device, in some future change.

Knowing that the device operates once every three minutes, for twenty-one uninterrupted hours, six days a week, it would suffer failure due to fatigue. Furthermore, due to the high voltage level, the drive rod is expected to withstand a relatively low number of cycles, to the order of 50,000 only, as shown in Figure 15.

Considering this brand, the drive rod will have a useful life of about 138 days, as shown in Figure 16, respecting the maximum wear limit. Figure 15 and Figure 16 were generated from MATLAB 5.1^®^ software.

In the structural analysis of the drive rod of the extraction device using AISI 420 stainless steel, the component presented a value of 216.07 MPa for maximum Von-Mises equivalent stress, being a stress almost three times smaller than the yield stress of the material used, as shown in Figure 17.

The most critical safety factor found was approximately 2.60, in the region of curvature of the rod. This value appears in a region with a small area and with a value well above unity, thus indicating that the component withstands the efforts well, under the conditions analyzed.

Unlike the prototype of the drive rod of the extraction device, whose analyses were performed using AISI 304 stainless steel, in this case, now using AISI 420 stainless steel, the drive rod showed an infinite life result (represented in the software by the value of 1 × 10^8^ cycles throughout the component body). Thus, the drive rod, in this case, would only fail due to random and external causes that are not predictable.

It is also worth noting that, considering 1 × 10^8^ cycles, as shown in Figure 18, is exactly the moment of component failure and considering the condition that the device operates once every three minutes and that the equipment the device is inserted in runs about twenty-one hours a day, the component lifetime would be approximately 238,095 days, which corresponds to about 650 years without interruption. Figure 18 was generated from MATLAB 5.1^®^ software.

In the structural analysis of the base of the extraction device made of AISI 304 stainless steel, the most critical safety factor found was approximately 3.19, in the region for the fixation of the mainspring in the base, as expected, due to the cut performed on the plate. This value appeared in a small region and was much larger than unity. This indicates that the component supports the efforts well.

The component presented a value of 64.001 MPa for maximum Von-Mises tension, where compared to the yield stress of the material used, it was almost four times smaller, as shown in Figure 19. However, with the use of AISI 420 stainless steel, it considerably increased this value in order to ensure greater robustness and reliability at that critical point of the component.

Considering the maximum allowable wear of half a millimeter (half the thickness of the plate that constitutes the base) and given the changes in the strategy of the production plan for the equipment in which the device is inserted, it was adopted that it works uninterruptedly twenty-one hours per day, six days a week, with each device performing its function once every three minutes.

This provides an estimated usage time of three months and 22 days, as shown in Figure 20, which represents a total of 4.172 × 10^4^ activation cycles, as shown in Figure 21.

In the structural analysis of the base of the extraction device made of AISI 420 stainless steel, the most critical safety factor found was approximately 8.44, also in the region intended for fixing the mainspring to the base, as expected. This value appeared in a small region as well as in the structural analysis of the base of the device made of AISI 304 stainless steel, having a value almost three times higher.

The component presented a value of 64.193 MPa for maximum tension of Von-Mises times, where compared to the yield stress of the material used, it was almost nine times smaller, as shown in Figure 22. With this, the expectation is that with the use of this material, there is a guarantee of greater robustness and reliability at this critical point of the component.

Considering the maximum allowable wear of half a millimeter (half the thickness of the plate that constitutes the base) and given the strategy of the production plan for the equipment in which the device is inserted, it was adopted that it works uninterruptedly twenty-one hours a day, six days a week, with each device performing its function once every three minutes.

This provides us with an estimated usage time of 1549 days (four years and 10 months), as shown in Figure 23, which represents a total of 6.508 × 10^5^ trigger cycles, as shown in Figure 24.

## 4. Conclusions

The use of AISI 420 stainless steel in the device components provided significant increases in their mechanical strength, ranging from 67% to 85%.There were considerable increases in the values of the respective safety factors associated with the use of AISI 420 stainless steel, ranging from 83% to 164%, and consequent increases in the useful life of the components, which ranged from 30% to 650%, when compared to the same performance parameters of the components where AISI 304 stainless steel was used.The improvement in these parameters guarantees, thus far, a replacement time (systematic time for preventive maintenance) of devices using AISI 420 stainless steel twice as long as the replacement time of extraction devices using stainless steel AISI 304, directly impacting the associated cost [33].The total cost of manufacturing the extraction device using AISI 420 stainless steel was around one third of the cost of the extraction device using AISI 304 stainless steel.

## Figures and Tables

**Figure 1 materials-15-00280-f001:**
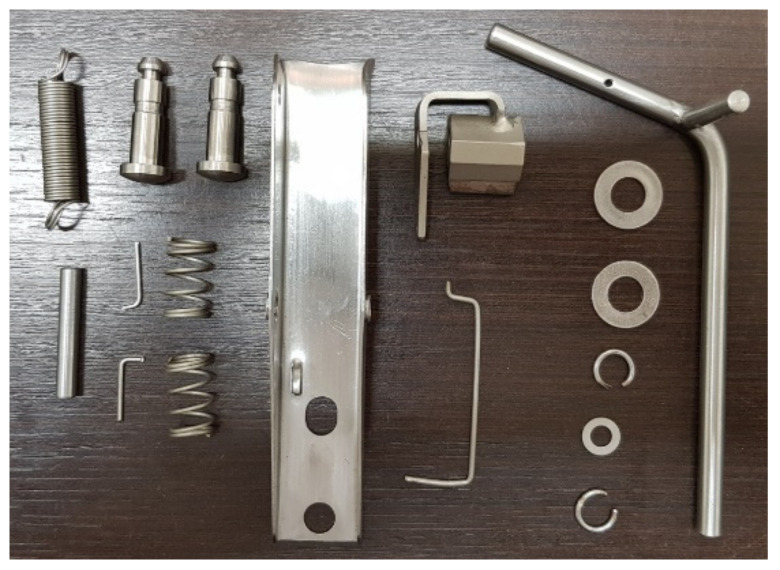
Elements selected from existing devices, intended for making specimens for the analysis and testing of materials.

**Figure 2 materials-15-00280-f002:**
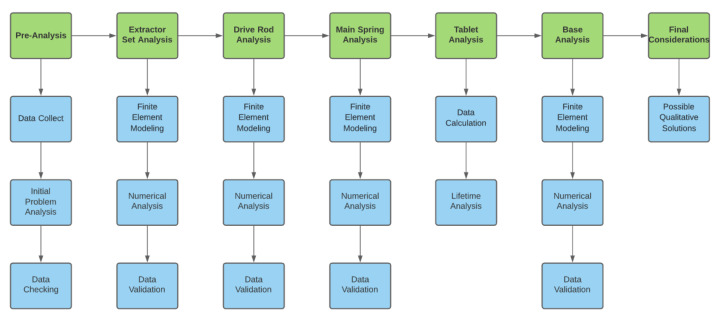
Systemic plan to support the execution of structural analysis activities.

**Figure 3 materials-15-00280-f003:**
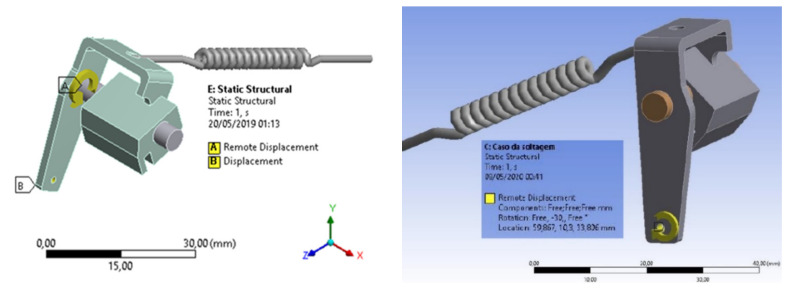
Structural analysis of the extractor set with defined boundary conditions.

**Figure 4 materials-15-00280-f004:**
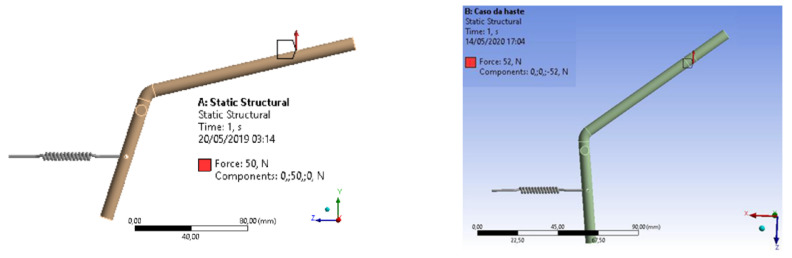
Structural analysis of the drive rod with defined boundary conditions.

**Figure 5 materials-15-00280-f005:**
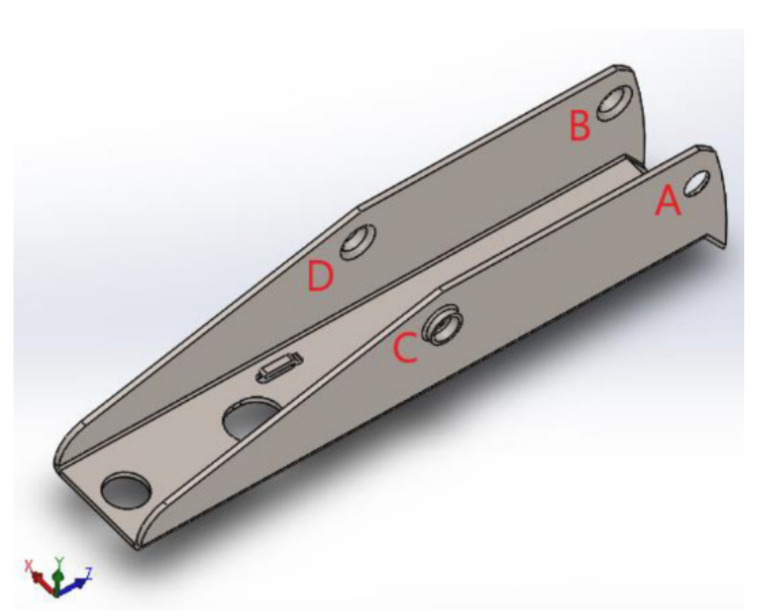
Indication of the holes and their representations, within the scope of the device base project.

**Figure 6 materials-15-00280-f006:**
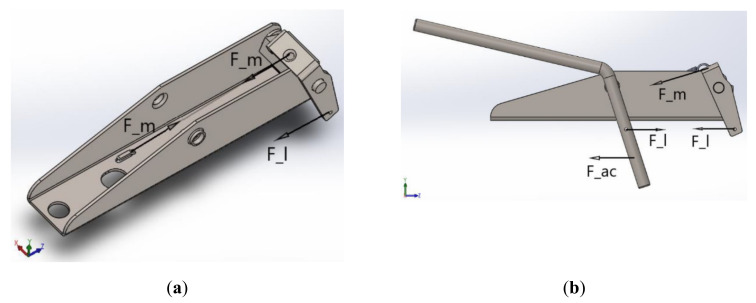
Representation of the forces acting in “Parts I and II”, within the scope of the device base design: (**a**) Part I; (**b**) Part II.

**Figure 7 materials-15-00280-f007:**
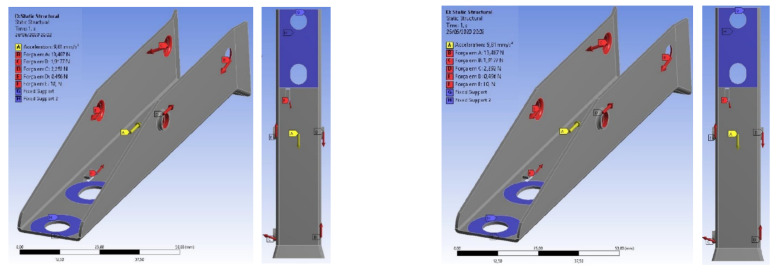
Structural analysis of the device base with defined boundary conditions.

**Figure 8 materials-15-00280-f008:**
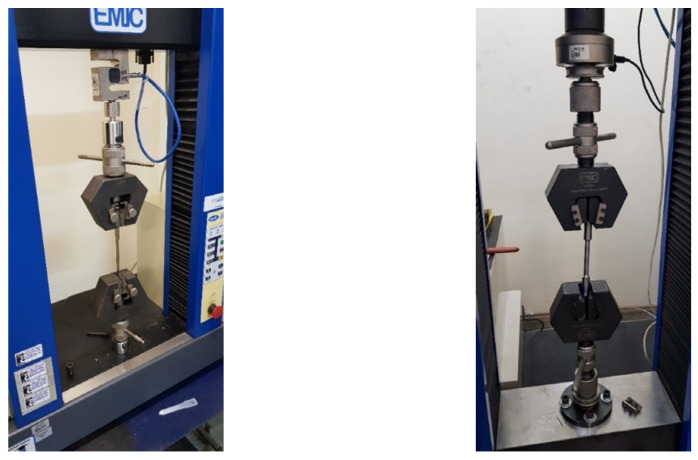
Execution of tensile tests on the analyzed materials.

**Figure 9 materials-15-00280-f009:**
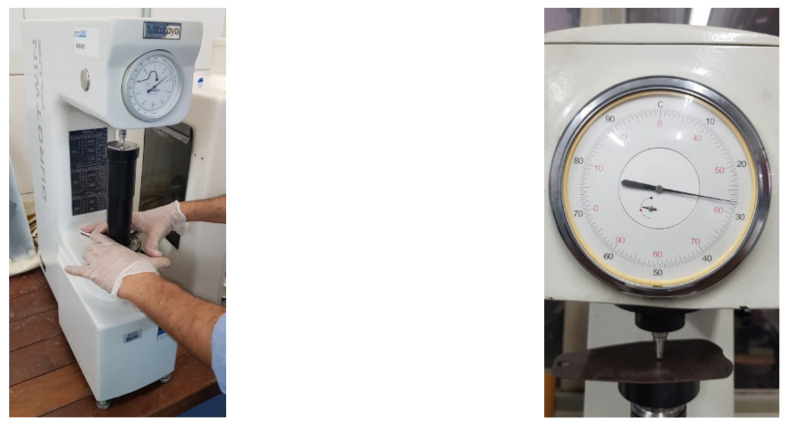
Carrying out hardness analysis of the analyzed materials.

**Figure 10 materials-15-00280-f010:**
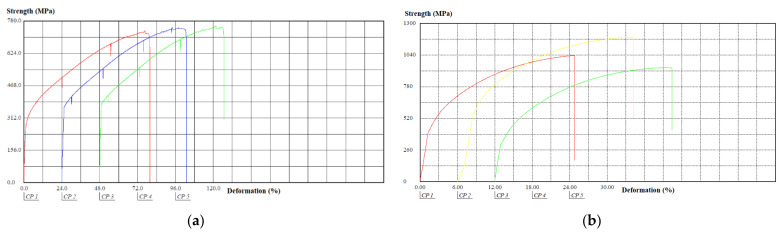
Behavior of tensile–deformation curves of components in sheet and cylindrical shapes, manufactured in AISI 304 stainless steel: (**a**) Sheet shape; (**b**) Cylindrical shape.

**Figure 11 materials-15-00280-f011:**
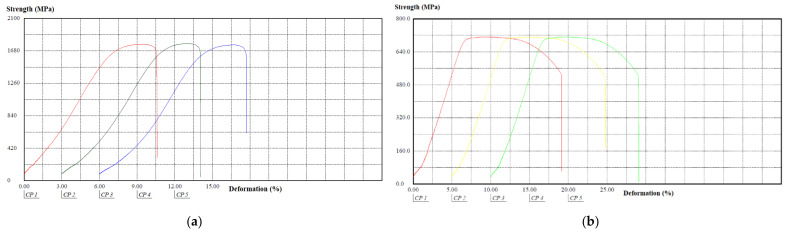
Behavior of tensile-deformation curves of components in sheet and cylindrical shapes, manufactured in AISI 420 stainless steel: (**a**) Sheet shape; (**b**) Cylindrical shape.

**Figure 12 materials-15-00280-f012:**
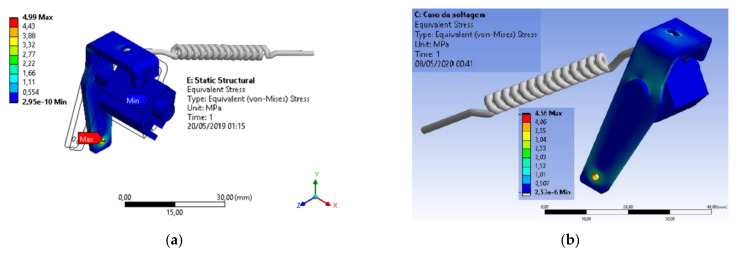
Structural analysis of the extractor assembly: (**a**) Component manufactured in AISI 304 stainless steel; (**b**) Component made of AISI 420 stainless steel.

**Figure 13 materials-15-00280-f013:**
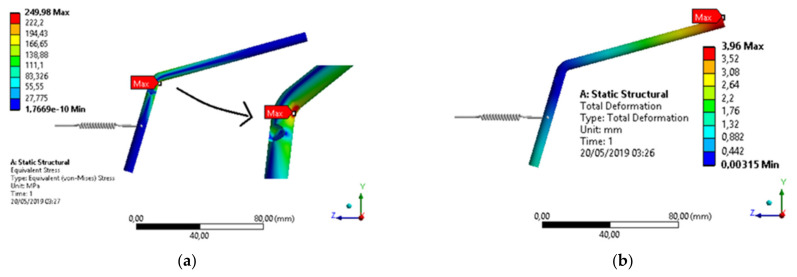
Structural analysis of the drive rod manufactured in AISI 304 stainless steel: (**a**) Stress distributions obtained by the Von-Mises criterion; (**b**) Deformations in the component, caused by stress applications.

**Figure 14 materials-15-00280-f014:**
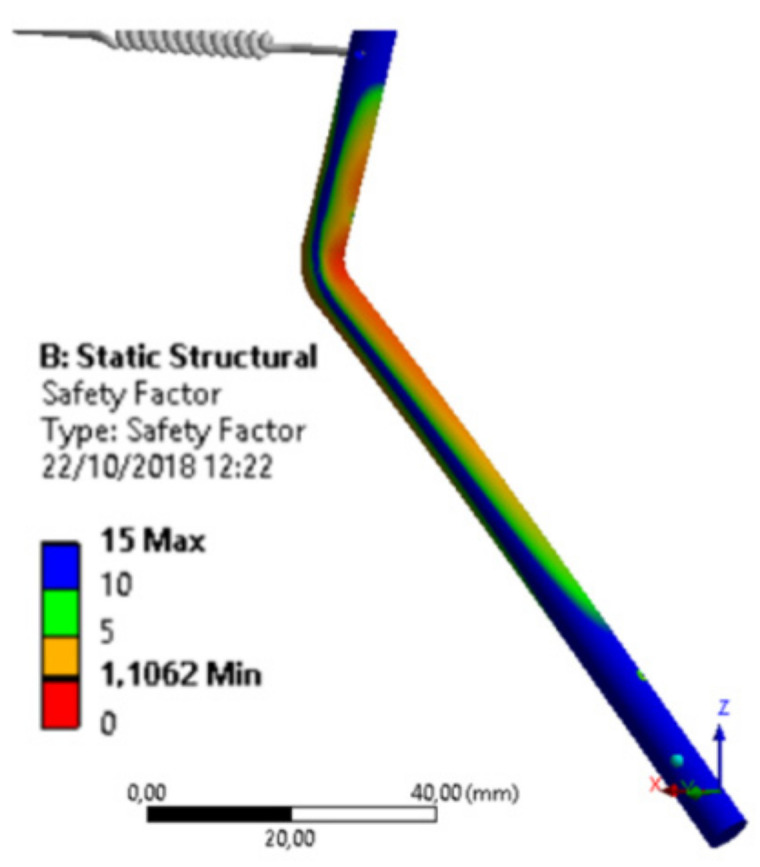
Safety factors along the drive rod structure made of AISI 304 stainless steel.

**Figure 15 materials-15-00280-f015:**
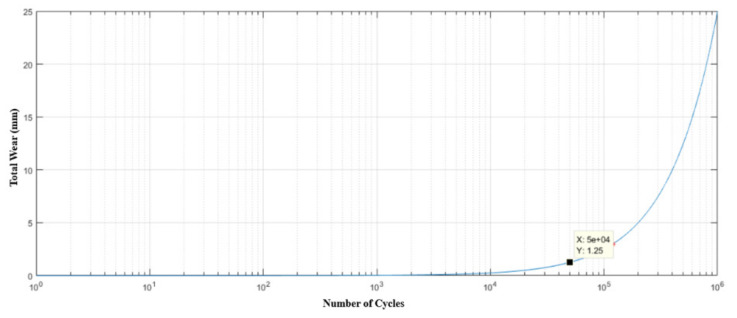
Life estimation curve (number of cycles) of the drive rod manufactured in AISI 304 stainless steel in relation to its wear.

**Figure 16 materials-15-00280-f016:**
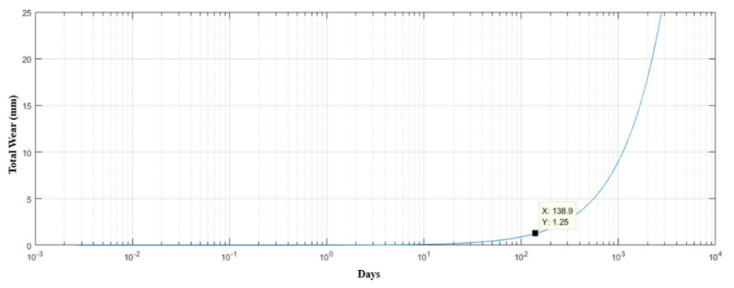
Life estimation curve (number of days) of the drive rod manufactured in AISI 304 stainless steel in relation to its wear.

**Figure 17 materials-15-00280-f017:**
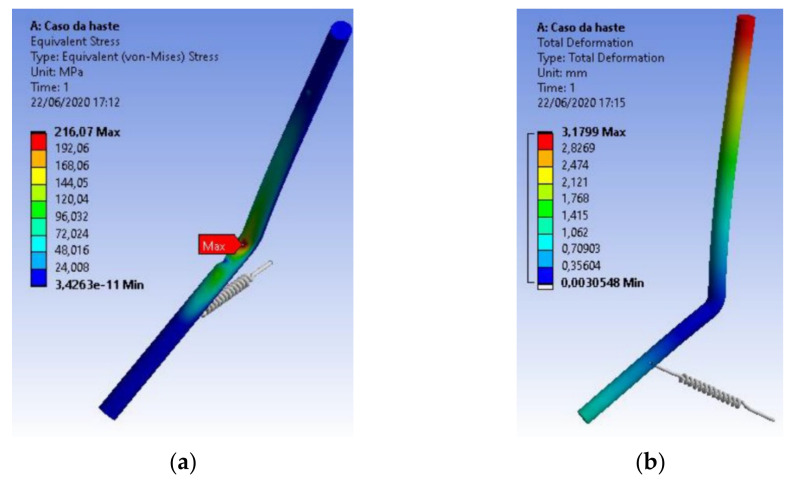
Structural analysis of the drive rod made of AISI 420 stainless steel: (**a**) Stress distributions obtained by the Von-Mises criterion; (**b**) Deformations in the component, caused by stress applications.

**Figure 18 materials-15-00280-f018:**
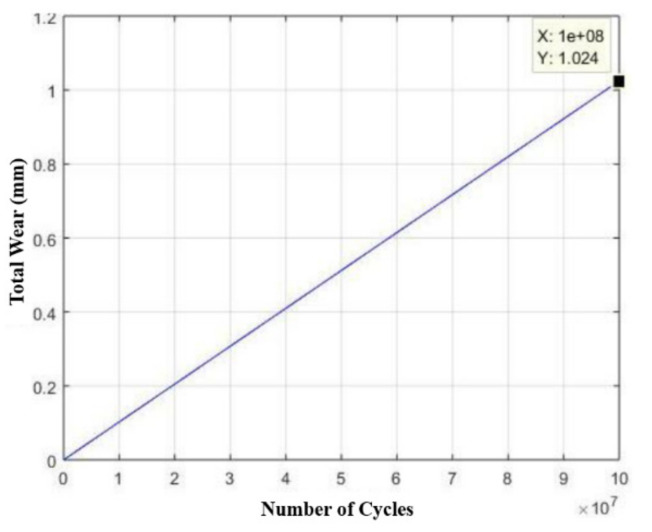
Life estimation curve (number of cycles) of the drive rod manufactured in AISI 420 stainless steel in relation to its wear, considering the equipment in which the device is inserted, works seven days a week.

**Figure 19 materials-15-00280-f019:**
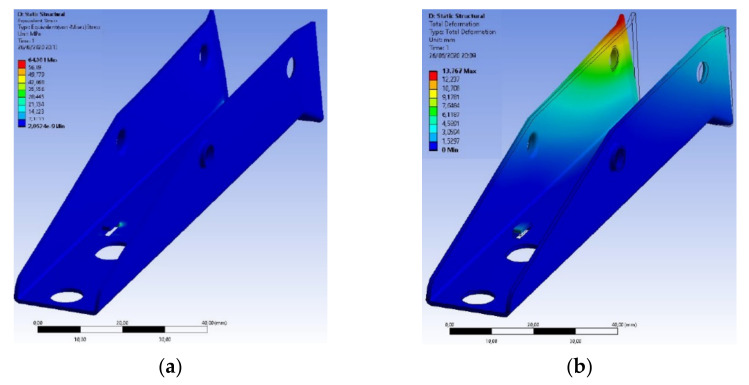
Structural analysis of the base made of AISI 304 stainless steel: (**a**) Stress distributions obtained by the Von-Mises criterion; (**b**) Deformations in the component caused by stress applications.

**Figure 20 materials-15-00280-f020:**
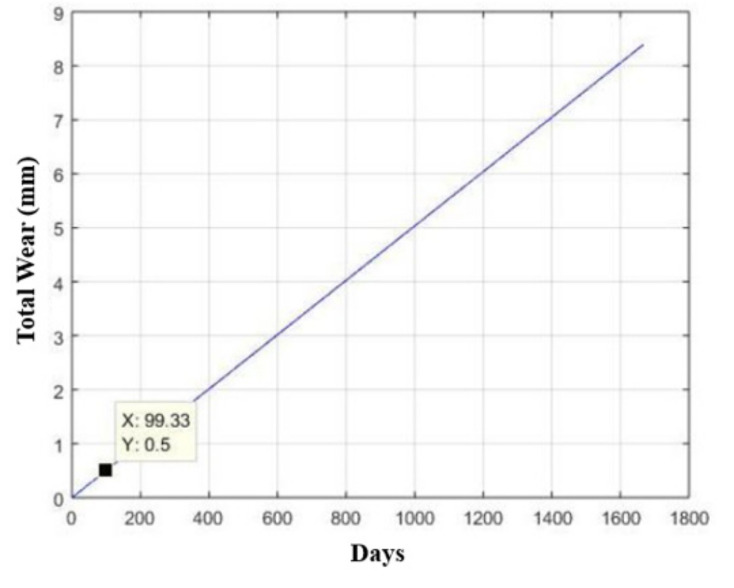
Curve for estimating the useful life (number of days) of the base made of AISI 304 stainless steel in relation to its wear, considering a maximum allowable wear of half a millimeter.

**Figure 21 materials-15-00280-f021:**
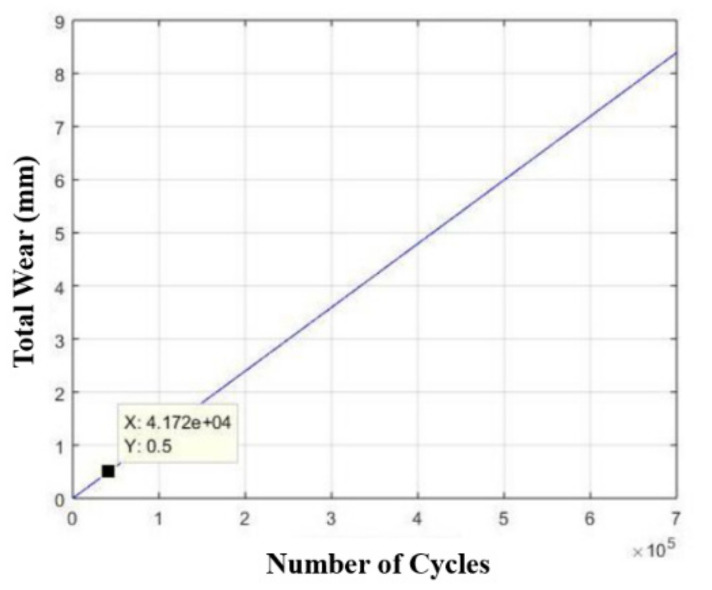
Estimated service life curve (number of cycles) of the base made of AISI 304 stainless steel in relation to its wear, considering a maximum allowable wear of half a millimeter.

**Figure 22 materials-15-00280-f022:**
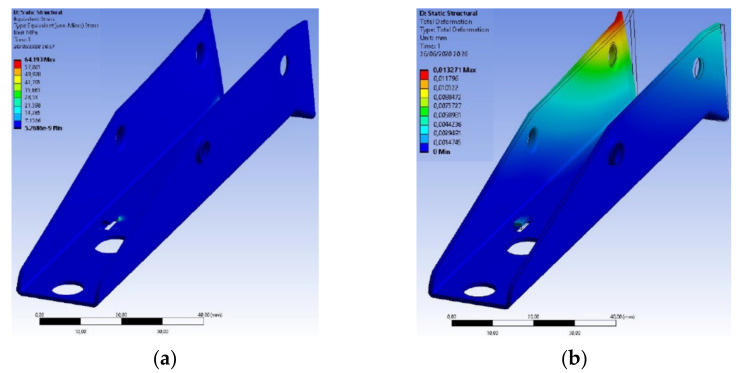
Structural analysis of the base made of AISI 420 stainless steel: (**a**) Stress distributions obtained by the Von-Mises criterion; (**b**) Deformations in the component caused by stress applications.

**Figure 23 materials-15-00280-f023:**
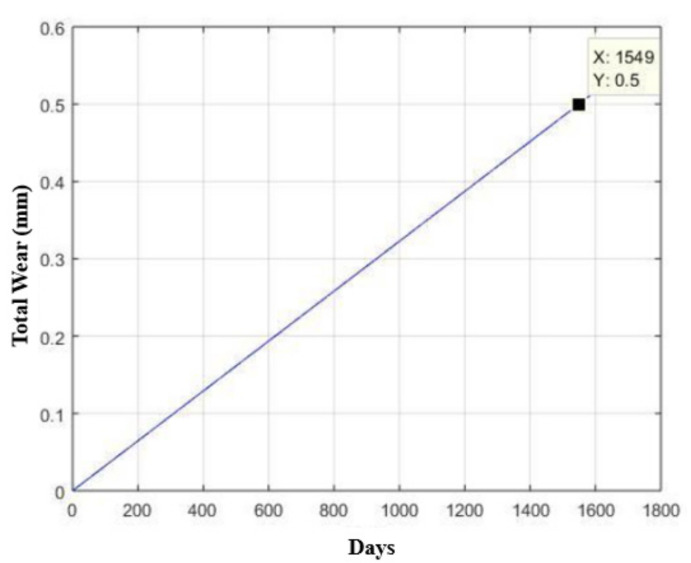
Curve for estimating the useful life (number of days) of the base made of AISI 420 stainless steel in relation to its wear, considering a maximum allowable wear of half a millimeter.

**Figure 24 materials-15-00280-f024:**
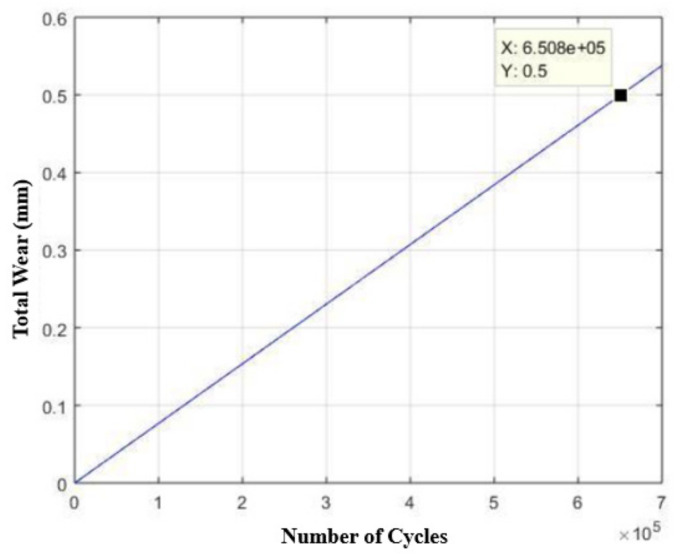
Estimated service life curve (number of cycles) of the base made of AISI 304 stainless steel in relation to its wear, considering a maximum allowable wear of half a millimeter.

**Table 1 materials-15-00280-t001:** Specifications of the materials used in the manufacture of prototypes in AISI 304 stainless steel and the final version of the device developed in AISI 420 stainless steel.

Specification of Materials Prototype	Specification of Materials Final Version of the Developed Device
ASTM A240 2B AISI 304L Steel Sheet 1.00 × 1300 × 2000 mm	AISI 420 Steel Sheet 1.00 × 300 × 1000 mm
ASTM A240 2B AISI 304L Steel Sheet 2.00 × 1300 × 2000 mm	AISI 420 Cylindrical Steel Bar Diameter 5 mm
AISI 304L Steel Bar Cylindrical Diameter 5 mm	AISI 420 Cylindrical Steel Bar Diameter 6 mm
AISI 304L Steel Bar Cylindrical Diameter 6 mm	AISI 420 Cylindrical Steel Bar Diameter 6.35 mm
AISI 304L Steel Bar Cylindrical Diameter 6.35 mm	AISI 420 Cylindrical Steel Bar Diameter 9.53 mm
AISI 304L Steel Bar Cylindrical Diameter 9.53 mm	AISI 420 Cylindrical Steel Bar Diameter 11.11 mm
AISI 304L Steel Bar Cylindrical Diameter 11.11 mm	AISI 420 Cylindrical Steel Bar Diameter 12.70 mm
AISI 304L Steel Bar Cylindrical Diameter 12.70 mm	AISI 420 Cylindrical Steel Bar Diameter 22.22 mm
AISI 304L Steel Bar Cylindrical Diameter 22.22 mm	AISI 420 Square Steel Bar Thickness 15.87 mm
AISI 304 Steel Bar Square Thickness 15.87 mm	

**Table 2 materials-15-00280-t002:** Results of the tensile tests performed on components manufactured in AISI 304 stainless steel.

Sample Format	Assessed Magnitude	
	Tensile Strength Limit (MPa)	Rupture Strain (MPa)
Plate 1	732.38	633.86
Plate 2	748.84	650.08
Plate 3	757.71	661.53
Cylinder 1	1038.34	1038.34
Cylinder 2	1189.78	1038.34
Cylinder 3	938.11	936.34

**Table 3 materials-15-00280-t003:** Results of the tensile tests performed on components manufactured in AISI 420 stainless steel.

Sample Format	Assessed Magnitude	
	Tensile Strength Limit (MPa)	Breaking Stress (MPa)
Plate 1	1768.12	1598.62
Plate 2	1777.49	1668.64
Plate 3	1759.67	1604.46
Cylinder 1	712.75	531.86
Cylinder 2	712.83	517.79
Cylinder 3	712.92	525.06

**Table 4 materials-15-00280-t004:** Results of hardness analyses performed on components manufactured in AISI 304 stainless steel.

Analysis Type	Sheet Shape		Cylindrical Shape	
	Average Hardness	Standard Deviation	Average Hardness	Standard Deviation
Vickers Hardness	198.66 HV	4.46	238.20 HV	5.34
Hardness Hockwell	11.00 HR	4.46	20.00 HR	5.34

**Table 5 materials-15-00280-t005:** Results of hardness analyses performed on components manufactured in AISI 420 stainless steel.

Analysis Type	Sheet Shape		Cylindrical Shape	
	Average Hardness	Standard Deviation	Average Hardness	Standard Deviation
Vickers Hardness	227.80 HV	5.83	271.00 HV	6.93
Hardness Hockwell	17.00 HR	5.83	26.00 HR	6.93

## Data Availability

Not applicable.

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
