# Peer review of "Study for Performance Increase of a Extractor Device by Steel Replacement of AISI 304 Steel for AISI 420 Steel"

_materials, 2021, doi:10.3390/ma15010280_

Round 1

Reviewer 1 Report

Notes:

There are no fundamental remarks on the content of the article. There is a note on inaccuracies in terminology and the sequence of presentation:

  1. Hardness HRC is erroneously referred to by the authors as Hockwell hardness. The real name of the scientist after whom this method is named H. M. Rockwell (in the designation HR is the Hardness of Rockwell).
  2. In the section "Materials and methods" in paragraph 2.2.2. it is necessary to give the parameters of the tensile tests and the standard according to which the tests were carried out. In paragraph 2.2.3. it is not necessary to describe the essence of HRC and HV hardness tests. It is enough to indicate the standard and test parameters. All material scientists know these methods. It is better to remove Figures 8 and 9 in this section, since they do not carry any important semantic meaning - the methods are standard.
  3. From clause 3.1. in the "Results and their discussion" section, the information on the rate of deformation of AISI 304 and AISI 420 steel samples during tensile tests should be transferred to paragraph 2.2.2.
  4. In the "Results and Discussion" section, the values of the tensile and hardness test results HRC and HV in Tables 2, 3, 4 and 5 must be given in whole numbers. Also, the values of standard deviations must be given in whole numbers.

Author Response

Response to Reviewer 1 Comments

Point 1.

Hardness HRC is erroneously referred to by the authors as Hockwell hardness. The real name of the scientist after whom this method is named H. M. Rockwell (in the designation HR is the Hardness of Rockwell).

Response 1: In the revised manuscript submitted, the Rockwell Hardness designation changed from HRC to HC.

Point 2.

In the section "Materials and methods" in paragraph 2.2.2. it is necessary to give the parameters of the tensile tests and the standard according to which the tests were carried out.

Response 2: In the revised manuscript submitted, in the section "Materials and methods" in paragraph 2.2.2. the parameters of the tensile tests were provided (constant strain rate of 0.0025, s-1) and the standard according to which the tests were carried out (Brazilian Regulatory Standard ABNT NBR 6892- Metallic Materials- Tensile Testing Test Method à Room temperature).

In paragraph 2.2.3. it is not necessary to describe the essence of HR and HV hardness tests. It is enough to indicate the standard and test parameters. All material scientists know these methods.

Response 2: In the revised manuscript submitted, in paragraph 2.2.3. as they are two different methods, the descriptions of the HR and HV hardness tests were kept. In addition, the standards and test parameters were indicated - ABNT NBR ISO 6508 standard (Metallic materials - Hockwell hardness test) and ABNT NBR NM ISO 6507 standard (Metallic materials - Vickers hardness test) respectively.

It is better to remove Figures 8 and 9 in this section, since they do not carry any important semantic meaning - the methods are standard.

Response 2: Even though all material scientists are aware of these methods, it is convenient to keep Figures 8 and 9 in this section, since students, especially from countries other than Brazil, may not know the respective methods listed, as they are governed by Brazilian standards.

Point 3.

From clause 3.1. in the "Results and their discussion" section, the information on the rate of deformation of AISI 304 and AISI 420 steel samples during tensile tests should be transferred to paragraph 2.2.2.

Response 3: In the revised manuscript submitted, in clause 3.1. in the section "Results and their discussion", information on the deformation rate of steel samples AISI 304 and AISI 420 during tensile tests has been transferred to paragraph 2.2.2.

Point 4.

In the "Results and Discussion" section, the values of the tensile and hardness test results HRC and HV in Tables 2, 3, 4 and 5 must be given in whole numbers. Also, the values of standard deviations must be given in whole numbers.

Response 4: In the "Results and Discussion" section, the values ​​of the results of the HR and HV tensile and hardness tests in Tables 2, 3, 4 and 5 will be kept accurate to two decimal places, due to the level of precision and reliability of the equipment used in the tests, as well as to be able to meet the requirements in the standards in reference, already mentioned in the previous item. Standard deviation values ​​will also be kept accurate to two decimal places, for the same reasons.

Reviewer 2 Report

The manuscript is aimed at modeling the extractor device operation process using ANSYS. It was declared that mechanical properties were taken into account, but it is not clear from the manuscript which mechanical properties were taken into account apart from the tensile strength limit and hardness. It is also not clear (Table 2) why breaking stress for cylinder 3 is higher than the tensile strength limit, although in other cases we can see the opposite trend (Table 2 and Table 3).

The extractor device operates under complex loading, i.e. impact and fatigue load and wear. However, corresponding experimental characteristics of the studied steels are not provided in the manuscript.

The manuscript presents a lot of results related to the modeling of various components of the extractor, but no materials science characteristics affecting the performance of this product are provided except the tensile strength limit and hardness characteristics.

Author Response

Response to Reviewer 2 Comments

Point 1.

It was declared that mechanical properties were taken into account, but it is not clear from the manuscript which mechanical properties were taken into account apart from the tensile strength limit and hardness.

Response 1: Which properties that were taken into account, in addition to the tensile strength and hardness limit, were:

1 - Rupture Strain (MPa)

2 - Wear resistance (mm) in relation to the service life in days

3 - Wear resistance (mm) in relation to the number of actuation cycles by the device

Point 2.

It is also not clear (Table 2) why breaking stress for cylinder 3 is higher than the tensile strength limit, although in other cases we can see the opposite trend (Table 2 and Table 3).

Response 2: In Table 2, the rupture stress value for cylinder 3 was entered incorrectly. The correct value is 936.34 (MPa), being changed in the corrected version of the manuscript sent.

Point 3.

The extractor device operates under complex loading, i.e. impact and fatigue load and wear. However, corresponding experimental characteristics of the studied steels are not provided in the manuscript.

Response 3: The experimental characteristics corresponding to complex loads, that is, impact load and fatigue, referring to the studied steels are given in the manuscript, particularly in section 3.3 (see figures 15, 16, 18, 20, 21, 23 and 24), through the correlations below:

1 - Steel wear resistance (mm) in relation to service life in days

2 - Steel wear resistance (mm) in relation to the number of activation cycles of the device

Point 4.

The manuscript presents a lot of results related to the modeling of various components of the extractor, but no materials science characteristics affecting the performance of this product are provided except the tensile strength limit and hardness characteristics.

Response 4: In addition to the tensile strength limit and hardness characteristics, several other material science characteristics affect the performance of this product, as listed below:

1 - Rupture Strain (MPa) - Tables 2 and 3

2 - Wear resistance (mm) in relation to the service life in days - Figures 15, 20 and 23

3 - Wear resistance (mm) in relation to the number of actuation cycles by the device - Figures 16, 18, 21 and 24

4 - Safety Factor - See Conclusions

Round 2

Reviewer 2 Report

  1. Rupture Strain (MPa) – MPa is the unit of stress. Strain cannot be measured in MPa.
  2. It is not clear how the wear was measur It follows from the manuscript that this is a calculated value based on the results of hardness measurements using ANSYS?
  3. The experimental characteristics corresponding to complex loads, that is, impact load and fatigue, referring to the studied steels are given in the manuscript, particularly in section 3.3 (see figures 15, 16, 18, 20, 21, 23 and 24), through the correlations below:

1 – Steel wear resistance (mm) in relation to service life in days

2 – Steel wear resistance (mm) in relation to the number of activation cycles of the device.

It is not clear what characteristics of fatigue or impact load are taken into account. If these are characteristics that are obtained experimentally, it is necessary to describe the method of obtaining them. Because it seems that all the results (except strength and hardness) are obtained by the calculation method. Besides, it is not clear how complex loads, impact load, and fatigue are taken into account.

The obtained results are very interesting from the engineering point of view, but from the point of view of materials science the answers to the above questions are required.

Author Response

Response to Reviewer 2 Comments - Round 2

Point 1.

Rupture Strain (MPa) – MPa is the unit of stress. Strain cannot be measured in MPa.

Response 1: All deformations are listed in % as shown in the graphics (Figures 10 and 11).

Point 2.

It is not clear how the wear was measur It follows from the manuscript that this is a calculated value based on the results of hardness measurements using ANSYS?

Response 2: The wear on the elements of the device was measured in an industrial plant, during their operation.

After checking the wear, the wear estimates per device operating cycle or per time in days were calculated by the Matlab application, as listed in the manuscript.

Point 3.

The experimental characteristics corresponding to complex loads, that is, impact load and fatigue, referring to the studied steels are given in the manuscript, particularly in section 3.3 (see figures 15, 16, 18, 20, 21, 23 and 24), through the correlations below:

1 – Steel wear resistance (mm) in relation to service life in days

2 – Steel wear resistance (mm) in relation to the number of activation cycles of the device.

It is not clear what characteristics of fatigue or impact load are taken into account. If these are characteristics that are obtained experimentally, it is necessary to describe the method of obtaining them. Because it seems that all the results (except strength and hardness) are obtained by the calculation method. Besides, it is not clear how complex loads, impact load, and fatigue are taken into account.

Response 3: The characteristics of mechanical resistance to wear, fatigue and impact were obtained from field measurements with the device in operation, and then extrapolated to the long term using Matlab software.

In Matlab, the constructive data obtained in the analyzes carried out in the Ansys software were considered.